# How well does the virtual format of oncology multidisciplinary team meetings work? An assessment of participants' perspectives and limitations: A scoping review

**Muhammad Abdul Rehman** [ID] *, **Unaiza Naeem** [ID], **Anooja Rani, Umm E. Salma Shabbar Banatwala, Afia Salman, Muhammad Abdullah Khalid, Areeba Ikram, Erfa Tahir**

Department of Internal Medicine, Dow Medical College, Dow University of Health Sciences, Karachi, Pakistan

* abdurehman528@gmail.com

## Abstract

### Background

Virtual multidisciplinary team meetings (VMDTM) provide a standard of care that is not limited by physical distance or social restrictions. And so, when the COVID-19 pandemic imposed irrefutable social restrictions and made in-person meetings impossible, many hospitals switched to the VMDTMs. Although the pandemic might have highlighted the ease of VMDTMs, these virtual meetings have existed over the past decade, albeit less in importance. Despite their recent importance, no review has previously assessed the feasibility of VMDTMs through the eyes of the participants, the barriers participants face, nor their comparison with the in-person format. We undertook this scoping review to map existing literature and assess the perspectives of VMDTM participants.

### Material and methods

We searched MEDLINE, Embase, CINAHL, and Google Scholar from inception till July 1st, 2023 to select studies that evaluated the perspectives of participants of VMDTMs regarding the core components that make up a VMDMT. Four authors, independently, extracted data from all included studies. Two authors separated data into major themes and sub-themes.

### Results

We identified six core, intrinsic aspects of a VMDTM that are essential to its structure: (1) organization, (2) case discussion and decision-making, (3) teamwork and communication, (4) training and education, (5) technology, and (6) patient-related aspect. VMDTMs have a high overall satisfaction rating amongst participants. The preference, however, is for a hybrid model of multidisciplinary teams. VMDTMs offer support to isolated physicians, help address complex cases, and offer information that may not be available elsewhere. The periodical nature of VMDTMs is appropriate for their consideration as CMEs. Adequate technology is paramount to the sustenance of the format.

**Data Availability Statement:** All relevant data are within the paper and its Supporting Information files.

**Funding:** The author(s) received no specific funding for this work.

**Competing interests:** All authors of this review work with, or have worked with, the not-for-profit organization, "Tumor Board Establishment Facilitation Forum (TEFF)". TEFF is a student-run organization that helps establish, and offers access to, multidisciplinary tumor boards at the tertiary care hospital, Dr. Ruth Pfau Civil Hospital in Karachi, Pakistan. This does not alter our adherence to PLOS ONE policies on sharing data and materials.

## Conclusion

VMDTMs are efficient and offer a multidisciplinary consensus without geographical limitations. Despite certain technical and social limitations, VMDTM participants are highly satisfied with the format, although the preference lies with a hybrid model.

## 1 Introduction

In oncology, the growing complexity of cancer care necessitates a collaborative approach to management. Tumor boards or oncology multidisciplinary teams have proven effective platforms in this respect, by marshaling the collective expertise of a diverse group of specialists, such as medical oncologists, surgical oncologists, radiation oncologists, pathologists, radiologists, and others [1]. Before the 1990s, cancer cases in the United Kingdom were primarily managed by general medical practitioners, with a limited number of patients receiving consultation from specialized oncologists, leading to a lamentable disparity in cancer care [2]. This radically changed after the Calman-Hine report was published in 1995, which established multidisciplinary team meetings (MDTM) as the cornerstone of cancer management [3].

With the rise of telemedicine, virtual MDTMs (VMDTM) have emerged as a promising solution for geographical barriers, allowing experts from different institutions to collaborate in the virtual sphere [4]. Though VMDTMs are not a new concept [5], they have expanded in popularity in recent years. The COVID-19 pandemic, in particular, served as an impetus for caregivers who sought ways to continually provide optimal patient care while adhering to social distancing [6, 7]. Many hospitals transitioned to VMDTMs, and this virtualization continues to gain traction [2, 8]. Therefore it is not illogical to state that the movement toward virtual tumor boards is an offshoot of the COVID-19 pandemic [6].

Studies integrating available literature to offer a broad map of VMDTMs are scarce. Prior reviews have evaluated VMDTMs in the context of specific cancer [4], and MDTMs in general without primarily focusing on the virtual format [9–11], or in regional oncology networks [12]. Thus, a comprehensive, all-encompassing study examining the perspectives of participating physicians themselves regarding VMDTMs, not limited to a particular geography, is notably absent from the existing literature. Furthermore, no study has previously assessed the effectiveness of VMDMTs in comparison to in-person MDTMs (IMDTMs).

Therefore, to map existing literature regarding VMDTM effectiveness through participant experience, from when it was first described, and through the COVID-19 pandemic to date, we conducted a scoping review of the literature. Scoping reviews are a relatively new form of systematic research to synthesize evidence, and the distinction from a systematic review is often confusing [13]. Scoping reviews map literature and provide a broad view of the topic by listing the volume of available literature, and study characteristics, and lay the groundwork for a more focused and contemporary systematic review [13, 14]. Scoping reviews often do not offer an assessment of quality or risk of bias for individual studies unless warranted by the primary objective [13, 15].

This knowledge gap is particularly salient given the increasing integration of VMDTMs in oncological practice, especially during the COVID-19 pandemic. And so, with this scoping review, we aim to fill the knowledge gap by exploring the following primary questions:

1. What are the perspectives harbored by VMDTM participants on the effectiveness of individual, core intrinsic components of a VMDTM?

2. What are the barriers and enabling factors for VMDTM participants?

3. What are participant perspectives regarding VMDTM core components in comparison to the in-person format?

## 2 Materials and methods

We conducted this scoping review according to the Preferred Reporting Items for Systematic reviews and Meta-Analyses extension for Scoping Reviews (PRISMA-ScR) guidelines [16]. The PRISMA-ScR checklist is present in **S1 File.**

### 2.1 Search strategy and study selection

We searched MEDLINE, Embase, CINAHL, and Google Scholar from inception till July 1st, 2023. The search string is listed in **S2 File**. A citation search was also performed by going through the reference lists of the short-listed articles to identify other relevant studies.

Studies were screened based on titles, abstracts, and full text by two authors (MAR and UN). Any difference in opinion was resolved through discussion between the authors (MAR and UN). If the difference in opinion persisted, another author (AR) of this review was consulted for clarification.

### 2.2 Inclusion/exclusion criteria

By the scoping nature of this review, we report all articles that assessed participants' opinions regarding any intrinsic aspect of VMDTMs. There was no limit on the study design to be eligible, except for case reports, which were deemed insufficient to evoke an accurate opinion regarding the workings of the VMDTM.

The inclusion criteria were satisfied by articles reporting all three of the following: (1) any intrinsic feature or aspect of VMDTMs, (2) VMDTMs used for oncological case discussions, and (3) an evaluation by the persons participating in the VMDTM. The exclusion criteria were (1) case reports, (2) non-oncological VMDTMs, and (3) studies that did not evaluate an intrinsic feature of VMDTMs. Studies that were excluded during full-text evaluation are listed in **S3 File**.

### 2.3 Data extraction

After studies were shortlisted for inclusion in this review, four authors (USB, AS, AK, AI), independently, read each paper. All data that was reported through feedback from VMDTM participants and relevant to the intrinsic dynamics of a VMDTM were extracted (e.g. the quality of case discussion). Where authors could not ascertain whether a particular result was relevant to the intrinsic workings of a VMDTM or not, a fifth author (MAR) was consulted. Where the fifth author (MAR) could also not judge whether the result was relevant to the workings of a VMDTM, the result was included in the study. Data reported through review of records, or conclusions drawn by authors in the discussion of papers was not extracted.

### 2.4 Theme and sub-themes

The studies included in this review made use of different methods and individual surveys to assess their populations which offered varying information after the data extraction process. The extracted data was assessed and owing to the varying study designs employed by the studies, data coding was deemed unfeasible. Therefore, we used inductive reasoning (MAR, UN) to broadly classify the extracted data into broadly encompassing themes and sub-themes.

## 3 Results

### 3.1 Study characteristics

We discovered a total of 36 papers using our search criteria. The number of studies at each stage of the screening process is shown in **Fig 1**.

Study designs of included studies were mostly observational cross-sectional studies ($n = 15$) [7, 17–30]. Other study designs were retrospective review of records with a cross-sectional survey ($n = 12$) [5, 31–41], meeting observations with cross-sectional survey ($n = 1$) [42], mixed-method design ($n = 3$) [43–45], embedded study design ($n = 1$) [46], randomized controlled trials ($n = 2$) [47, 48], descriptive qualitative synthesis using free-text cross-sectional surveys ($n = 1$) [49], and an anthropological analysis ($n = 1$) [50]. A detailed summary of study characteristics is shown in **Table 1**. Further description of populations in included studies is described in **S4 File**.

A total of 10 studies evaluated feedback from VMDTMs during COVID-19 [7, 18, 19, 21–23, 25, 43, 44] or after the end of the pandemic [17]. There were 18 studies published between 2023 and 2020 [7, 17–27, 31–33, 43, 44, 49], 4 between 2019 and 2015 [28, 34, 42, 45], 6 studies between 2014 and 2010 [35–38, 40, 46], 4 studies between 2009 and 2005 [39, 41, 47, 48], and 3 between 2004 and 2000 [29, 30, 50]. One study was published in 1999 [5]. Most studies had populations from VMDTMs based in the United States of America ($n = 10$) [5, 18, 21, 25, 26, 33, 35, 37, 38, 46], followed by the United Kingdom ($n = 8$) [7, 17, 22, 23, 43, 44, 47, 48]. The distribution of these studies is described in the **S5 File**.

### 3.2 Theme identification

Across the range of studies included in this review, we identified six major themes that define the intrinsic mechanisms of a VMDTM: (1) organization, (2) case discussion and decision-making, (3) teamwork and communication, (4) education and training, (5) technology, and (6) patient-centered aspect. A thematic segregation of results from each study is shown in **Table 2**. Additional findings in the included studies are shown in **S6 File**.

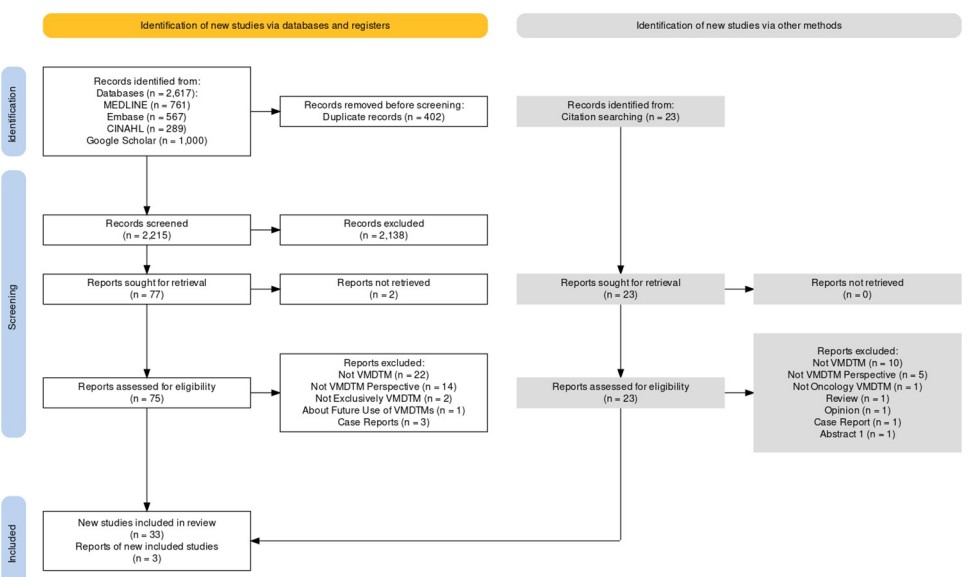

**Fig 1. PRISMA flowchart.** This figure shows the number of studies retrieved at each stage of the search strategy.

**Table 1. Characteristics of included studies.**

| # | Author/Year | Country/Region | Sample Size (RR) | Study Characteristics | Themes Evaluated |
|---|---|---|---|---|---|
| 1 | Ali SR *et al.* 2023 | UK | 68 (55.3%) | *Study Design*: Observational<br>*Data Collection Method*: Cross-sectional survey<br>*Response Measure*: 5-Point Likert Scale, MCQs, free-text responses<br>*Population*: VMDTM participants since COVID-19<br>*Time Period*: April 2022 | O, CD, TC, ET, T, P |
| 2 | Amin NB *et al.* 2023 | USA | 50 (56%) | *Study Design*: Observational<br>*Data Collection Method*: Cross-Sectional Survey<br>*Response Measure*: 5-Point Likert Scale, MCQs<br>*Population*: VMDTM participants during COVID-19<br>*Time Period*: August 2021 to October 2021 | CD, TC, ET, P |
| 3 | Groothuizen JE *et al.* 2023 | UK | Interviews, 56*; survey, 73 (67.7%) | *Study Design*: Mixed-method study<br>*Data Collection Method*: Interviews, cross-sectional surveys, meeting observations<br>*Response Measure*: Free-text responses<br>*Population*: VMDTM participants during COVID-19<br>*Time Period*: April 2021 to August 2021 | O, CD, TC, T |
| 4 | Soukup T *et al.* 2023 | UK | 423 (NR) | *Study Design*: Mixed-method<br>*Data Collection Method*: Cross-sectional survey<br>*Response Measure*: Free-text responses, sliding scale marker of 0–100<br>*Population*: VMDTM participants during COVID-19<br>*Time Period*: September 2020 to August 2021 | O, T |
| 5 | Thiagarajan S *et al.* 2023 | India | 52 (NR) | *Study Design*: Observational<br>*Data Collection Method*: Cross-sectional survey<br>*Response Measure*: Close-ended MCQs<br>*Population*: VMDTMs centres<br>*Time Period*: December 2016 to February 2022 | CD, ET, P |
| 6 | Bonanno N *et al.* 2023 | Many | 204 (NR) | *Study Design*: Observational<br>*Data Collection Method*: Cross-sectional survey<br>*Response Measure*: MCQs<br>*Population*: VMDTM radiologists during COVID-19<br>*Time Period*: September 2021 | O, CD, TC, T |
| 7 | Esteso F *et al.* 2022 | Argentina | NR | *Study Design*: Observational<br>*Data Collection Method*: Review of records, cross-sectional survey<br>*Response Measure*: NR<br>*Population*: VMDTM participants<br>*Time Period*: NR | – |
| 8 | Perlmutter B *et al.* 2022 | USA | 253** (23.3%) | *Study Design*: Observational<br>*Data Collection Method*: Cross-sectional survey<br>*Response Measure*: Modified anchored Likert Scale (for comparison, -10 to 10; for problems and suggestions, 0 to 10)<br>*Population*: VMDTM participants during COVID-19<br>*Time Period*: September 2020 to October 2020 | O, CD, TC, ET, T |
| 9 | Thallinger C *et al.* 2022 | Central and Southeastern Europe | 14 (NR) | *Study Design*: Observational<br>*Data Collection Method*: Review of records, cross-sectional surveys, knowledge questionnaires<br>*Response Measure*: NR<br>*Population*: VMDTM participants<br>*Time Period*: June 2018 to July 2021 | – |
| 10 | Cathcart P *et al.* 2021 | UK | 18 (NR) | *Study Design*: Observational<br>*Data Collection Method*: Cross-sectional surveys<br>*Response Measure*: NR<br>*Population*: VMDTM participants during COVID-19<br>*Time Period*: NR | TC, T |

*(Continued)*

**Table 1.** (Continued)

| # | Author/Year | Country/Region | Sample Size (RR) | Study Characteristics | Themes Evaluated |
|---|---|---|---|---|---|
| 11 | Mohamedbhai *et al.* 2021 | UK | 97 (NR) | *Study Design*: Observational<br>*Data Collection Method*: Cross-sectional surveys<br>*Response Measure*: NR<br>*Population*: VMDTM participants during COVID-19<br>*Time Period*: December 2020 to February 2021 | O, CD, TC, ET, T, P |
| 12 | Rajasekaran RB *et al.* 2021 | UK | 36 (92.3%) | *Study Design*: Observational<br>*Data Collection Method*: Cross-sectional surveys<br>*Response Measure*: NR<br>*Population*: VMDTM participants during COVID-19<br>*Time Period*: May 2020 | O, CD, TC, T |
| 13 | Rosabal-Obando M *et al.* 2021 | Latin America | 103 (72.5%)*** | *Study Design*: Observational<br>*Data Collection Method*: Cross-sectional surveys<br>*Response Measure*: 5-Point Likert Scale<br>*Population*: VMDTM participants<br>*Time Period*: October 2017 to October 2018 | CD |
| 14 | Dharmarajan H *et al.* 2020 | USA | 19 (NR) | *Study Design*: Observational<br>*Data Collection Method*: Cross-sectional surveys<br>*Response Measure*: NR<br>*Population*: VMDTM participants during COVID-19<br>*Time Period*: NR | O |
| 15 | Habermann TM *et al.* 2020 | USA | NR | *Study Design*: Observational<br>*Data Collection Method*: Review of records, cross-sectional survey<br>*Response Measure*: 5-Point Likert Scale<br>*Population*: VMDTM participants<br>*Time Period*: January 2014 to June 2018 | CD, ET, T |
| 16 | Pan M *et al.* 2020 | USA | Survey 1 (2013), 6 (NR); Survey 2, (2015), 32 (NR); Survey 3 (2019), 54 (NR) | *Study Design*: Observational<br>*Data Collection Method*: Cross-sectional surveys<br>*Response Measure*: NR<br>*Population*: VMDTM participants<br>*Time Period*: March 2013 to December 2019 | ET |
| 17 | Rosell L *et al.* 2020 | Sweden | 125 (52%) | *Study Design*: Descriptive qualitative<br>*Data Collection Method*: Cross-sectional survey<br>*Response Measure*: Free-text answers<br>*Population*: VMDTM participants<br>*Time Period*: May 2017 to May 2018 | O, CD, TC, ET, T, P |
| 18 | Sidpra J *et al.* 2020 | UK, USA | 24 (48%) | *Study Design*: Observational<br>*Data Collection Method*: Cross-sectional survey<br>*Response Measure*: NR<br>*Population*: VMDTM participants during COVID-19<br>*Time Period*: NR | O, CD, TC, ET, T, P |
| 19 | Abu Arja MH *et al.* 2019 | Latin America | 95 (60%) | *Study Design*: Descriptive Cross-sectional<br>*Data Collection Method*: Cross-sectional survey<br>*Response Measure*: NR<br>*Population*: VMDTM participants<br>*Time Period*: December 2017 to January 2018 | O, CD, ET, T |
| 20 | Rosell L *et al.* 2019 | Sweden | 125 (52%) | *Study Design*: Observational<br>*Data Collection Method*: Cross-sectional survey, Observation<br>*Response Measure*: 7-Point Likert Scale (respondents), 5-Point Likert Scale (MODe and MOT)<br>*Population*: VMDTM participants<br>*Time Period*: May 2017 to May 2018 | O, CD, TC, ET, T, P |

(*Continued*)

**Table 1.** (Continued)

| # | Author/Year | Country/Region | Sample Size (RR) | Study Characteristics | Themes Evaluated |
|---|---|---|---|---|---|
| 21 | Van Huizen LS *et al.* 2019 | Netherlands | 6 (NA) | *Study Design*: Mixed-method study<br>*Data Collection Method*: Review of records, semi-structured interviews<br>*Response Measure*: Theme identification<br>*Population*: VMDTM participants<br>*Time Period*: September 2016 to February 2017 | CD |
| 22 | Crispen C *et al.* 2015 | The Bahamas, Trinidad and Tobago | 10 (100%) | *Study Design*: Observational study<br>*Data Collection Method*: Review of records, cross-sectional survey<br>*Response Measure*: 5-Point Likert Scale<br>*Population*: VMDTM participants<br>*Time Period*: July 2013 to November 2013 | O, CD, ET, T |
| 23 | Marshall CL *et al.* 2014 | USA | 36 (100%) | *Study Design*: Prospective observational study<br>*Data Collection Method*: Review of records, cross-sectional survey<br>*Response Measure*: 5-Point Likert Scale<br>*Population*: VMDTM participants<br>*Time Period*: Four months | O, CD, TC, ET, T |
| 24 | Shea CM *et al.* 2014 | USA | Interviews, 28 (NA); Cross-sectional surveys, 32 (100%) | *Study Design*: Embedded case study<br>*Data Collection Method*: Interviews, cross-sectional surveys, observations<br>*Response Measure*: Theme identification, binary (yes/no) questions<br>*Population*: VMDTM participants<br>*Time Period*: August 2011 to October 2012 | O, CD, ET, T, P |
| 25 | Bold RJ *et al.* 2013 | USA | 412 (NA)**** | *Study Design*: Observational study<br>*Data Collection Method*: Review of records, cross-sectional survey<br>*Response Measure*: NR<br>*Population*: VMDTM participants<br>*Time Period*: September 2008 to December 2009 | ET, P |
| 26 | Stevenson MM *et al.* 2013 | USA | 10 (50%) | *Study Design*: Observational study<br>*Data Collection Method*: Review of records, cross-sectional survey<br>*Response Measure*: NR<br>*Population*: VMDTM participants<br>*Time Period*: December 2009 to 2012 | O, CD, TC |
| 27 | Donnem T *et al.* 2012 | Norway | 141 (84.4%) | *Study Design*: Prospective observational study<br>*Data Collection Method*: Review of records, cross-sectional survey<br>*Response Measure*: MCQs<br>*Population*: VMDTM participants<br>*Time Period*: March 2009 to September 2010 | ET, T, P |
| 28 | Schroeder JK *et al.* 2011 | Germany | 205 (74.4%) | *Study Design*: Observational study<br>*Data Collection Method*: Review of records, cross-sectional survey<br>*Response Measure*: MCQs<br>*Population*: VMDTM participants<br>*Time Period*: December 2004 to June 2009 | O, CD, TC, ET |
| 29 | Chekerov R *et al.* 2008 | Germany | Survey 1, 43 (57%); Survey 2, 51 (68%) | *Study Design*: Observational study<br>*Data Collection Method*: Review of records, cross-sectional survey<br>*Response Measure*: MCQs<br>*Population*: VMDTM participants<br>*Time Period*: December 2004 to August 2006 | O, CD, ET, T |

(*Continued*)

**Table 1.** (Continued)

| # | Author/Year | Country/Region | Sample Size (RR) | Study Characteristics | Themes Evaluated |
|---|---|---|---|---|---|
| 30 | Kunkler IH *et al.* 2007 | UK | VMDTM vs. IMDTM: 1867 (86%) vs. 882 (88%) | *Study Design*: Randomized Controlled Trial<br>*Data Collection Method*: Review of records, cross-sectional surveys<br>*Response Measure*: 5-Point Likert Scale<br>*Population*: VMDTM and IMDTM participants<br>*Time Period*: March 2004 to April 2005 | CD, TC |
| 31 | Savage SA *et al.* 2007 | Scotland | 50 (58.8%) | *Study Design*: Observational study<br>*Data Collection Method*: Review of records, cross-sectional survey<br>*Response Measure*: 5-Point Likert Scale<br>*Population*: VMDTM participants<br>*Time Period*: NR | O, CD, TC, ET, T, P |
| 32 | Kunkler I *et al.* 2006 | UK | VMDTM vs. IMDTM: 24 (72%) vs. 33 (73%) | *Study Design*: Randomized Controlled Trial<br>*Data Collection Method*: Cross-sectional surveys sent before the trial and during week 28<br>*Response Measure*: GBI scores ranging from 0–12 or 0–16<br>*Population*: VMDTM and IMDTM participants<br>*Time Period*: March 2004 to September 2004 | CD, TC, T |
| 33 | Delaney G *et al.* 2004 | Australia | VMDTM vs. IMDTM: 16 (61.5%) vs. 16 (59%) | *Study Design*: Observational study<br>*Data Collection Method*: Anthropological study, pre-trial and post-trial cross-sectional surveys<br>*Response Measure*: Anthropological assessment, MCQs<br>*Population*: VMDTM and IMDTM participants<br>*Time Period*: February 2000 to June 2000 | CD, TC, T |
| 34 | Gagliardi *et al.* 2003 | Canada | 67 (51.1%) | *Study Design*: Observational study (Pilot Study)<br>*Data Collection Method*: Needs' assessment, cross-sectional surveys<br>*Response Measure*: 5-Point Likert Scale, MCQs<br>*Population*: VMDTM participants<br>*Time Period*: October 2001 to March 2002 | O, CD, TC, ET, T |
| 35 | Olver IN *et al.* 2000 | Australia | 20 (100%) | *Study Design*: Observational<br>*Data Collection Method*: Cross-sectional surveys<br>*Response Measure*: MCQs<br>*Population*: VMDTM participants<br>*Time Period*: NR | O, CD, TC, ET, T, P |
| 36 | Hunter *et al.* 1999 | USA | 38 (100%) | *Study Design*: Observational<br>*Data Collection Method*: Review of records, cross-sectional surveys<br>*Response Measure*: 4-Point Likert Scale<br>*Population*: VMDTM participants<br>*Time Period*: March 1998 | O, CD, T |
| | | USA | 22 (38%) | *Study Design*: Observational<br>*Data Collection Method*: Review of records, cross-sectional surveys<br>*Response Measure*: 4-Point Likert Scale<br>*Population*: VMDTM participants<br>*Time Period*: March 1998 | |

*15 participants were interviewed twice

**333 responses but 253 completed the survey and were included in the study

***A total of 53 physicians were asked to fill multiple surveys over the study period

****A total of 28 physicians were asked to fill multiple surveys over the study period

**Abbreviations**: CD, case discussion; COVID-19, coronavirus disease; ET, education and training; GBI, Group Behaviour Inventory; IMDTM, in-person multidisciplinary team meeting; MODe, metric of decision-making; MOT, meeting observational tool; NA, not applicable; NR, not reported; O, organization; P, patient-centered aspect; MCQ, multiple-choice question; T, technology; TC, teamwork and communication; UK, United Kingdom; USA, United States of America; VMDTM, virtual multidisciplinary team meeting.

**Table 2. Theme-wise segregation of findings in included studies.**

| Study | Overall value and preference | Organization | Case Discussions and Decision-making | Teamwork and Communication | Education and Training | Technology | Patient-centered aspect |
|---|---|---|---|---|---|---|---|
| Ali SR *et al.* 2023 | *Preferred format*: IMDTM with the option to attend virtually, 87%; VMDTM, 8.7%; IMDTM, 4.3%. | [C] *Chairing and recording outcomes*: similar, 46.4%; difficult, 29%. [C] *Time-efficiency and organization*: better, 47.8%; same, 34.8%. [C] *Data validation and audit*: same, 46.4%. [D] VMDTM training would make meetings smoother. [D] Clinicians should prepare cases for discussion in MDTMs. | [C] *Decision-making*: similar, 75.4%. | [C] *Communication*: same, 47.8%; better, 29%; worse, 21.7%; uncertain, 1.4%. [C] *Inter-personal relationships and teamwork*: worse, 50.7% [C] *Engagement amongst specialties*: better, 37.7%; same, 37.7%; worse, 21.7%. [D] Non-verbal cues should be enhanced such as hands-up alerts to notify chairperson that a participant wants to communicate. [D] New participants should be integrated into the team by meeting in-person. | [C] *Training*; same, 52.2%; worse, 27.5%; better, 18.8%. [D] Allot time at the end of MDTMs for educational purposes. | [C] *Quality of imaging*: same, 37.7%; better, 37.7%; worse, 23.2%. [D] Better connectivity and the availability of technical support would be helpful. | [C] *Clinical trial recruitment*: same, 48.5%. [C] *Data security and patient confidentiality*: same, 72.5%. |
| Amin NB *et al.* 2023 | | | *Provide patient-specific information that impacts management*: agree, 96%; neutral, 2%; disagree, 2%; radiation/medical oncologist vs. surgeons, 85% vs 55%. *Useful in discussing complex cases with other clinicians*: agree, 98%; neutral, 0%; disagree, 2%. | *Better communication*: radiation/medical oncologist vs. surgeons, 85% vs 64%. *Improved access to other specialties*: agree, 84%; neutral, 10%; disagree, 4%; not specified, 2%; radiation/medical oncologist vs. surgeons, 63% vs 55%. | *Should be used for CMEs*: agree, 92%; neutral, 8%; disagree, 0%; academic vs community physicians, 64% vs 55%; academic oncologists vs community oncologists, 80% vs 50%. | | *Allow patients to stay updated about guidelines/trials*: agree, 82%; neutral, 14%; disagree, 2%; not specified, 2%; academic oncologists vs community oncologists, 60% vs 41%. *Allow access to clinical trial recruitment*: agree, 66%; neutral, 30%; disagree, 2%; not specified, 2%; academic vs community physicians, 64% vs 29%; academic oncologists vs community oncologists, 80% vs 23%. |
| Groothuizen JE *et al.* 2023 | [D] *Preferred format*: IMDTMs, but retain some part of VMDTMs. | [D] Solve the issue of travelling between centers. [D] Facilitate attendance. [D] Are time-efficient. [D] Coordinators cannot share their screens as the software allows only one screen to be visible. This caused the inability to correct errors/omissions. | [D] Better ease of accessing information. [D] Easier to share visual information. | [D] Participants can feel isolated. [D] Negatively impact informal communication during meetings. [D] *New participants may never see other participants because cameras may be off. [D] Cannot appreciate nuances and non-verbal impressions into account. [D] Improve meeting etiquette as participants do not talk over each other. *Participants who meet in-person outside the VMDTM reported fewer concerns regarding social limitations. | | [D] There are several technological issues. Poor connectivity is the main issue. [D] Poor technology compromised case-discussions. [D] Technology is the key factor for sustainability of VMDTMs. | |

*(Continued)*

**Table 2.** (Continued)

| Study | Overall value and preference | Organization | Case Discussions and Decision-making | Teamwork and Communication | Education and Training | Technology | Patient-centered aspect |
|---|---|---|---|---|---|---|---|
| Soukup T *et al.* 2023 | [D] *Preferred format*: Will maintain a hybrid nature post-pandemic because it allows more attendance, 85% (258/304). | [D] More clinicians are available for the MDT meeting which increased attendance, 41% (103/252). | | | | [D] IT-related issues are present and they cause delays causing the meeting to last longer, 43% (125/288). | |
| Thiagarajan S *et al.* 2023 | | | *Agreed with decision*, 96.1%. | | *Useful as an educational platform*, 96.1%. | | *Patients informed about decision*: yes, 50%; usually, 36.5%; sometimes, 13.5%. |
| Bonanno N *et al.* 2023 | [C] *Overall satisfaction*: same standard, 34.8; lower standard, 34.3%; improved standard, 30.9%. *Opinion regarding VMDTMs post-pandemic*: happy to attend, 69.6%; would not support, 30.4%. *Preferred format post-pandemic*: FtF, 35.8%; VMDTMs, 3.4%; mixed, 60.8%. | *Time spent preparing/participating*: 1–2 hours/week, 53.9%; 2–4 hours/week, 29.9%; >4 hours/week, 16.2%. *Chairing is more organized\**: yes, 26%. *Discussion is more organized*: yes, 22.1%. *Solve issues of scheduling and room availability*: yes, 47.1%. *Easier access*: yes, 57.4%. *Attendance of radiologists*: increased, 39.2%; maybe increased, 22.3%; same/decreased, 39.2%. *Attendance of non-radiologists*: increased, 40.2%. \*Because less people speak at one time. | *Discussion depth*: happy, 63.2%. | *Satisfactory interaction*: yes, 80.9%. *Lack of non-verbal cues cause misunderstandings*: yes, 48%. *Communication was ineffective between radiologists and other specialties*: yes, 23.5%. *Difficult to maintain working relationships with new members*: yes, 55.9%. | | *Accessing imaging data*: without issues, 77.5%. [C] *Viewing images*: equal or better, 70.1%; suboptimal/problematic, 29.4%. *Major obstacles*: difficulty with technology and connectivity, 67.2%; difficulty viewing images, 33.3%; lack of technical support, 43.1%. | |
| Esteso F *et al.* 2022 | *Overall satisfaction*: highly satisfied. 83%. | | | | | | |
| Perlmutter B *et al.* 2022 | *Overall satisfaction*: *Would you prefer VMDTMs if there was no pandemic*: yes, 46%. *Would you prefer VMDTMs if social restrictions were in place*: yes, 63.1%. *Despite modifications VMDTMs aren't as good as IMDTMs*: agree, 50.6%; strongly disagree, 36.7%; neutral, 12.7%. | [C] *Ease of presentation*: median score, 0. [P] *Could not see who is speaking*: median score, 3. *Multiple people speaking at once*: median score, 3; significantly more for trainees vs staff, *p* = 0.03. | [C] *Comprehensiveness of recommendations*: median score, 0. [C] *Quality of case discussion*: median score, -1. | [C] *Ease of getting sub-specialty recommendations*: median score, 0. [C] *Ease of getting multiple opinions*: median score, 0. *No opportunity to network*: median score, 5; significantly more for surgeons vs others, *p* = 0.004. | [C] *Ease of getting CME*: median score, 0. | [C] *Ease of reviewing pathology*: median score, 0. [C] *Ease of reviewing images*: median score, 0; median score for trainees vs staff, 5 vs 0, *p* = 0.03. [P] *Connectivity issues in VMDTM*: median score, 2; significantly more for physicians vs nurses, p = 0.02. [P] *Could not hear well*: median score, 3. [P] *Could not see presentation*: median score, 2. | |

(*Continued*)

**Table 2.** (Continued)

| Study | Overall value and preference | Organization | Case Discussions and Decision-making | Teamwork and Communication | Education and Training | Technology | Patient-centered aspect |
|---|---|---|---|---|---|---|---|
| Thallinger C et al. 2022 | *Quality*: excellent, 57%; good, 36%; average, 7%. | | | | | | |
| Cathcart P et al. 2021 *Approximate value from figure. | *Overall satisfaction*\*: high, 11; neither high nor low, 7; low, 0. [C] A majority considered VMDTM more efficient than IMDTM. | | | *Satisfaction with group interaction*\*: high, 10; neither high nor low, 3; low, 3. [D] Majority were confident to comment during VMDTM. Around 3/18 found it easier to comment in VMDTM vs. IMDTM. | | *Satisfaction with audio quality*\*: high, 7 (); neither high nor low, 7; low, 2. *Satisfaction with video quality*\*: high, 9; neither high nor low, 5; low, 1. [D] Participants encountered Initial short-term difficulties that were gone once members became accustomed to the format. | |
| Mohamedbhai H et al. 2021 | *Preferred format*: IMDTM with option to join virtually, 72.2%; IMDTM, 11.45%. | [C] *Time-efficiency and organization*: good, 42.3%; not better, 40.2%. [C] *Chairing and recording*: difficult, 38.1%; easy, 16.5%. | [C] *Decision-making*: unchanged, 70.1%; worse\*, 19.6%. \*70% were consultants | [C] *Interpersonal relationships and teamwork*: deteriorated, 69.1%. [C] *Engagement of participants*: worse, 43.9%; improved, 18.2%. [C] *Communication*: worse, 58.8%. | [C] *Training*: worse\*, 47.7%; better, 13.4%; same, 30.4%. \*70% of junior trainees rated worse. | *Technology*: satisfactory, 84.5%. [C] *Viewing case images/histology*: better, 41.2%; similar, 35.1%. | [C] *Data security/ patient confidentiality*: unchanged, 79.4%. |
| Rajasekaran RB et al. 2021 *Approximate value from figure. | *Overall satisfaction*: satisfied, 75%; neutral, 11.1%; dissatisfied, 13.9%. *Preferred format*: VMDTMs, 55.6%; want to participate in VMDTMs occasionally, 33.3%. | [C] *Time for case discussion is adequate*\*, 92%. | [C] *Adequate depth of discussion*: yes, 72.2%. [C] *Adequate number of specialties present*: yes, 88.9%. [C] *Change in treatment and decision-making remained unaffected*: yes, 100%. [C] *Conclusion to an appropriate diagnosis*\*: yes, 82%. | [C] *Adequate availability of specialists is possible*: yes, 88.9%. [C] *Adequate interaction between specialists*: yes, 72.2%. | | [C] *VMDTMs allowed access to all patient information i.e. images and slides*: yes, 86%. *Technical difficulty as the reason for not being a part in the future*: yes, 44%. | |
| Rosabal-Obando M et al. 2021 *Approximate value from figure. | *Overall satisfaction*: will recommend VMDTM: agree, 100%. | | *Recommendations were helpful*: agree, 100%\*. | | | | |
| Dharmarajan H et al. 2020 | *Preference*: VMDTM, 57.9%; IMDTM, 26.3%; no preference, 15.8%. *Preferred format post-pandemic*: VMDTMs, 78.9%; IMDTMs, 21.1%. | *Ease of format*: VMDTM easer, 31.6%; IMDTM easier, 31.6%; same ease, 36.8%. | | | | | |
| Habermann TM et al. 2020 | *Overall satisfaction*: overall activity is excellent, 100%. | | *Meeting objectives were met*: yes, 100%. | | *Appropriate references and teaching slides were included*: yes, 100%. *Saw improvement in knowledge and competence*: yes, 93%. [D] Meetings should have CME credits. | [D] Meetings should have real-time radiology images. | |

*(Continued)*

**Table 2.** (Continued)

| Study | Overall value and preference | Organization | Case Discussions and Decision-making | Teamwork and Communication | Education and Training | Technology | Patient-centered aspect |
|---|---|---|---|---|---|---|---|
| Pan M *et al.* 2020 | *Overall satisfaction*: improved quality of care: survey 1 (2015), 100%; survey 2 (2019), 94.4%. Median rating of 10/10 for helping physicians and patients (both surveys). | | | | *Improve confidence*: yes, 100% (both surveys). | | |
| Rosell L *et al.* 2020 | [D] *Overall satisfaction*: VMDTMs contribute to equity in care and better patient safety. | [D] Time-consuming and resource demanding because of preparatory work. [D] Coordinating center should document to enhance transparency and allocate time for evaluation and feedback. [D] Attendance is suboptimal, especially due to lack of resources in pathology, oncology and radiology; and irregular meeting dates. [D] There is a need for clarification of roles. [D] Difficulty in discussing all patients due to obscure referral principals, prestige, case overload and the need to select complex cases. | [D] Preparedness for meetings varies which negatively impacts quality of case discussion. [D] Feasible for discussion of complex cases. Participants were confident in the VMDTMs' expertise. [D] Discussions varied from structured to disorganized, with unresolved conflicts and the stress about performing at a national level. [D] Lack of relevant information caused deviation from recommendations given. [D] Allow national consensus and adherence to standards of care. [D] Lower commitment and preparedness for referred cases. | [D] Good for professional networking as the gap between different geographical areas is decreased. [D] Feeling of competition between centers can cause misunderstandings. | [D] Advantageous for low-volume centers as they expose them to numerous cases. [D] Discussions enhance individual and team competence. [D] Adequate for knowledge-sharing. [D] Nationwide referrals expose physicians to many cases that increase experience. | [D] Bad connectivity caused inability to see other participants or case-related information causing misunderstandings. [D] Transfer of radiology imaging between centers was time-consuming. | [D] Limited consideration of patients' comorbidity, care needs, performance status, and patients' perspectives. [D] Limited focus on clinical trials but have potential for clinical research collaborations. |
| Sidpra J *et al.* 2020 *Approximate value from figure. | [C] *Overall satisfaction*: standard of care: equal, 83.3%. | [C] *Accessibility*: equal, 81%*. [C] *Organization and running*: equal, 67%*. [C] *Environment*: equal, 74%*. [C] *Data collection and audit of outcomes*: equal, 64%*. | [C] *Clinical governance*: equal, 63%*. [C] *Clinical decision process and consensus*: equal, 67%*. | [C] *Communication*: equal, 67%*. **42% prefer IMDTMs because of better communication. | [C] *Education and professional development*: equal, 66%*. | [C] *Viewing images and pathology*: equal, 71%*; equal or better, 91.7%. | [C] *Security and patient confidentiality**: equal, 68%*. |

(*Continued*)

**Table 2.** (Continued)

| Study | Overall value and preference | Organization | Case Discussions and Decision-making | Teamwork and Communication | Education and Training | Technology | Patient-centered aspect |
|---|---|---|---|---|---|---|---|
| Abu Arja MH et al. 2019 | | *Frequency was adequate (n = 84)*: totally agree, 92%. *Duration was adequate (n = 84)*: totally agree, 83%. *Adequate time to prepare cases (n = 84)*: almost always, 79%. *Barriers to attendance\* (n = 95)*: workload, 64%; timing, 38%; problems accessing software and internet, 29%; incompatibility of recommendation with local unit, 2.4%; minimizes role of neuropathologists and pathologists from local region (Latin America), 1.2%; language barrier, 1.2%; lack of knowledge about time and day, 2.4%. *Using native language made discussion comprehensible and clear (n = 84)*: totally agree, 95; no, 2%. *Used post-meeting notes to stay updated (n = 95)*: almost always, 83%. *10.5% of respondents saw no barriers to attendance. | *Recommendations fit local circumstances (n = 84)*: almost always, 85%; occasionally/rarely, 15%. *Recommendations were clear to follow (n = 84)*: almost always, 99%. *Recommendations helped improve outcomes or quality of life of patients (n = 84)*: almost always, 96%. *Reviewed second opinion of pathology (n = 84)*: almost always, 57%; rarely/almost never, 31%. | | *Useful for continued medical education (n = 84)*: totally agree, 99%. | *Sending pathology slides was easy (n = 84)*: totally agree, 39%; neutral, 30%; totally disagree, 31%. | |
| Rosell L et al. 2019 | | *My role is clear*: agree, 85%; neutral, 2%; disagree, 13%. *Evaluate working with the VMDTM*: agree, 55%; neutral, 15%; disagree, 30%. *Leadership functions well\**: agree, 77%; neutral, 4%; disagree, 18%. *Guidelines for documentation of decisions are clear*: agree, 70%; neutral, 7%; disagree, 24%. \*In agreement with MDT-MOT and MDT-MODe | *Availability of case histories*: agree, 78%, neutral, 8%; disagree, 14%. | *Felt involved in the discussion\**: agree, 78%; neutral, 7%; disagree, 15%. *Were called upon to contribute to the meeting*: agree, 71%; neutral, 6%; disagree, 23%. \*In agreement with MDT-MOT and MDT-MODe | *Builds team competence\**: agree, 81%; neutral, 6%; disagree, 12%. *Builds individual competence\**: agree, 80%; neutral, 5%; disagree, 15%. \*In agreement with MDT-MOT and MDT-MODe | *Technology functioned well\**: agree, 63%; neutral, 14%; disagree, 22%. \*Unlike MDT-MOT and MDT-MODe which observed an even better functionality of technology. | *Patient comorbidity is considered\**: agree, 79%; neutral, 10%; disagree, 12%. *Patient's perspective is considered\**: agree, 71%; neutral, 11%; disagree, 18%. *Clear guidelines are present to inform patients about VMDTM decisions*: agree, 67%; neutral, 12%; disagree, 20%. \*Unlike MDT-MOT and MDT-MODe which observed weak consideration |
| van Huizen LS et al. 2019 | *Preference*: [D] Merge VMDT with site MDT. | | [D] Use in discussing complex cases by fine-tuning management plans. [D] Adds little value for routine cases that have established guidelines and may even irritate participants. [D] Discussing all cases is of no value and is outdated. | | | | |

*(Continued)*

**Table 2.** (Continued)

| Study | Overall value and preference | Organization | Case Discussions and Decision-making | Teamwork and Communication | Education and Training | Technology | Patient-centered aspect |
|---|---|---|---|---|---|---|---|
| Crispen C *et al.* 2015 | | *Number of cases that should be discussed per meeting*: no opinion, 89%. [D] *Difference of opinion regarding time of meeting.* | *Increase accountability*: agree, 100%; mean score, 3 ± 0. | | *Encourages research interest*: agree, 89%; disagree, 11%; mean score, 2.89 ± 0.78. | *Quality of audio/ visual was satisfactory*: agree, 100%; mean score, 3.4 ± 0.53. | |
| Marshall CL *et al.* 2014 | *Overall satisfaction*: 4.6 ± 0.5. | *Presentation time was sufficient*: 4.7 ± 0.5. *Discussion time was sufficient*: 4.7 ±0.5. | *Consensus was reached*: 4.6 ± 0.5. *Presentation and discussion was clear*: 4.6 ± 0.5. *Members had adequate knowledge of the topic*: 4.6 ± 0.5. *Decision was in best interest of patient*: 4.6 ± 0.5. | *Interaction with other participants was possible*: 4.7 ± 0.5. *Discussion and decision-making were adequately shared*: 4.6 ± 0.5. *Presenter built good rapport with audience*: 4.6 ± 0.6. *Presenter encouraged participation*: 4.6 ± 0.5. | *Stimulated critical thinking*: 4.6 ± 0.5. *Gave useful tips for personal practice*: 4.3 ± 0.8. *Caused reflection on my practice*: 4.3 ± 0.9. *Information was not available elsewhere*: 4.4 ± 0.8. *Event will change my current practice*: 3.8 ± 1.1. *Discussion was relevant to practice*: 4.6 ± 0.6. | *Visibility of participant*: 4.6 ± 0.7. *Visibility of slides*: 4.7 ± 0.6 *Quality of audio*: 4.5 ± 0.8. | |
| Shea CM *et al.* 2014 | *Overall satisfaction*: is an effective use of resources, 91% (no, 3%); is a valuable platform to discuss complex cases with community physicians, 100%. | *Will improve by*: streamlining pathology/radiology review will improve VMDTM, 44%; clear presentation guidelines, 31%; routinely collecting feedback from community-based physicians and communicating it to UNC participants, 28%; marketing VMDTM through campaign to community-based physicians, 59%; ensuring adequate representation of specialist physician expertise, 19%. [D] *Organizational barriers for community-based physicians*: (1) timing of the VMDTM, (2) time lag between scheduling and the presentation is not timely for patient management, (3) time required to prepare for a case presentation, (4) lack of reimbursement to presenting a case. | *Valuable in discussing complex cases with community physicians*: yes, 100; no, 0%. *Sufficient patient information for productive case discussion*: yes. 91%; no, 3%. *Cases by community-based physicians were appropriately complex for discussion*: yes, 97%. *Discussion focused on appropriate issues raised by presenter*: yes, 91%; no, 3%. [C] *VMDTM discussions reach consensus less frequently than IMDTMs*: yes, 16%; no, 66%. [D] *Obtaining a second opinion about treatment plans was the key reason for case presentation by community-based physicians.* | | [D] *Time spent reviewing literature by community physicians for their case inadvertently served to increase their clinical knowledge.* | *Technology problems frequently disrupt discussion*: yes, 9%; no, 88%. [D] *Firewall and bandwidth down speed were the main technological issues.* | *Valuable in recruiting patients for clinical trials*: yes, 44%; no, 16%. [D] *VMDTMs serve as an alternative for patients unwilling to travel to the designated hospital.* |
| Bold RJ *et al.* 2013 | *Overall satisfaction*: want to attend more VMDTMs, range, 85.1%-94.8%. | | | | *Provide new information that impacts patient care*: range, 90.3%-95.8%. *Impact physician practice regarding management of other cancer case*: range, 78.5%-85.8%. *Information regarding clinical trials in VMDTMs is satisfactory*: range, 94.8%-99.2%. | | *Allow familiarity with clinical trials*: range, 81.0%-93.1%. *More likely to enroll patients in clinical trials after VMDTM*: range, 67.0%-77.4%. |

*(Continued)*

**Table 2.** (Continued)

| Study | Overall value and preference | Organization | Case Discussions and Decision-making | Teamwork and Communication | Education and Training | Technology | Patient-centered aspect |
|---|---|---|---|---|---|---|---|
| Stevenson MM *et al.* 2013 | *Overall satisfaction*: will continue to participate in VMDTM, 70%. *Preference*: VMDTM, 40%; neutral, 40%; IMDTM, 20%. | *Day and time was convenient*, 60%. *Duration was adequate*, 100%. [D] Clash with other commitments was the most common reason for unwillingness to participate. | *Aided patient care especially due to radiology and pathology review*, 90%. | *Improved communication and referral times between sub-specialists*, 70%. | | | |
| Donnem T *et al.* 2012 | *Overall satisfaction*: it will improve patient care: very much, 65%; quite a bit, 31%; little, 3%; will not improve, 1%. | | | | *Impact on physician's confidence that patient care is adequate*: very much, 65%; quite, 31%; little, 3%; not at all, 1%. | *Technical problems*: very much, 3%; quite, 10%; little, 14%; not at all, 77%. | *Impact on patient's confidence that patient care is adequate*: very much, 65%; quite, 23%; little, 11%; not at all, 1%. |
| Schroeder JK *et al.* 2011 | *Overall satisfaction*: expectations fulfilled, 78.16%; expectations exceeded, 17.24%; expectations not fulfilled, 4.6%. | *Criteria for presenting a case*: rare cancer, 83.53%; ambiguous data, 56.47%. | *Cause reduction in patient management redundancies*: agree, 69.1%; disagree, 30.89%. *Asked for more second opinions*: yes, 50.57%. [C] Added-value in VMDTM compared to IMDTM: more information about patient, 53.34%; better quality of patient information, 53.34%; more material about clinical studies, 97.73%; better interdisciplinary of patients, 68.18%; better qualified participants, 93.33%; more clinical experience of participants, 79.07%. | *Coordination between care sectors*: improved, 70%. A significant number of outpatient participants agreed compared to inpatient participants ($p = 0.002$). Reasons: faster (87.72%) and detailed exchange (66.67%) of knowledge. | *Participants' knowledge increased*, 80.68%. | | |
| Chekerov R *et al.* 2008 | | *Time saving benefit*: large, 72%; moderate, 20%; no, 8%. *Moderation unit was good*, 84%. | *Methodical and didactic assemblage of the suggested topics*, 96%. *Relevant case selection*, 92%. *Optimal to support difficult clinical decisions*, 92%. | | *Have extensive scientific discussion*, 92%. *Optimal for advanced educational training*, 81%. *Beneficial for clinical practice*, 88%. | *Technical support*: good, 72%; satisfying, 28%. *Software easy to operate*, 80%. *Design of web interface was good*, 84%. | |
| Kunkler IH *et al.* 2007 | | | [C] *Consensus was reached*: 92% vs. 97%; neutral, 6% vs. 3%; disagree, <3% vs. <1%; $p = 0.048$. [C] *Decision was in best interest of patient*: agree, 93% vs. 96%; neutral, 6% vs. 4%; disagree, <1% vs. <1%. | [C] *Discussion was adequately shared between participants*: agree, 91% vs. 95%; neutral, 6% vs. 4%; disagree, 3% vs. 1%. | | | |

*(Continued)*

**Table 2.** (Continued)

| Study | Overall value and preference | Organization | Case Discussions and Decision-making | Teamwork and Communication | Education and Training | Technology | Patient-centered aspect |
|---|---|---|---|---|---|---|---|
| Savage SA *et al.* 2006 | | *Timing is appropriate*, 92%. *Frequency is appropriate*, 96%. *Ratification of final treatment plan*: good, 75%; fair, 18%; poor, 6%. | *Patient management*: good, 77%; fair, 22%. *Case background clinical information*: good, 88%; fair, 12%. *Case pathology information*: good, 90%; fair, 10%. *Case radiology information*: good, 74%; fair, 20%; poor, 6%. *Images/video clips*: good, 61%; fair, 27%; poor, 10%. | *Discussion with other specialties*: good, 67%; fair, 31%; poor, 2%. | *As an educational platform*: good, 88%; fair, 10%; poor, 2%. | | *Patient identification for clinical trials*: poor, 38%; good, 33%; fair, 29%. |
| Kunkler I *et al.* 2006 *Out of 12 **Out of 16 | [C] *Overall satisfaction**: 10.3 vs. 10.8. | | [C] *Efficient decision-making**: 10.8 vs 11.5. [C] *Satisfactory meeting output**: 10.5 vs. 11.5. [C] *Provision of knowledge resources (data, imaging, professionals)**: 7.9 vs 8.9. | [C] *Discussion behavior***: 9.4 vs. 9.4. [C] *Discussion norms***: 11.0 vs. 11.3. [C] *Meeting climate**: 8.5 vs. 8.3. [C] *Level of shared goals***: 8.5 vs. 8.4. [C] *Group status**: 7.5 vs. 8.7. | | [C] *Provision of physical resources (data, imaging, technical equipment)**: 8.5 vs. 7.3. | |
| Delaney G *et al.* 2004 | [C] *Overall satisfaction*: good quality, 68.8% vs. 68.8%; satisfactory quality, 31.3% vs. 6.3%; poor quality, 0% vs. 0%; did not attend, 0% vs. 25%. *Preference*: IMDTM, 68.7%; VMDTM, 31.3%. | | *Attendance aids case management*: yes, 87.5% (50% said that it aids significantly); no, 6.3%; not applicable, 6.3%. *Radiology was useful*: yes, 87.6%; not useful, 12.5%. *Pathology was useful*: yes, 63.1%; not useful, 37.5%. [D, C] Scientific evidence used vs. multiples ways of knowing. | [D, C] Centre was dominant vs. equality among participants [D, C] Within-specialization resolution vs. resolution across specialties. [D, C] Uncertainty was not acknowledged vs. uncertainty was acknowledged [D, C] No joking vs. jokes present [D, C] Close-ended communication vs. open-ended communication. | | *Quality of audio*: sufficient, 50%; inadequate, 37.5%; really Irritating, 12.5%. *Quality of video*: sufficient, 68.7%; inadequate, 31.3%; really Irritating, 0%. | |
| Gagliardi *et al.* 2003 | *Overall satisfaction*: satisfied, 74.6%; neutral, 20.9%; not satisfied, 4.5%. | *Sufficient presentation time*: agree, 83.5%; neutral, 17.9%; disagree, 3.0%. *Sufficient discussion time*: agree, 74.6%; neutral, 22.4%; disagree, 3%. [D] Be observant of "raised hands". | *Clear presentation of topic*: agree, 80.6%; neutral, 17.9%; disagree, 1.5%. *Presenter had good knowledge*: agree, 94%; neutral, 4.5%; disagree, 1.5%. | *Possible to interact with speaker*: agree, 77.6%; neutral, 20.9%; disagree, 1.5%. *Possible to interact with other participants*: agree, 59.7%; neutral, 17.9%; disagree, 16.4%; no response, 6.0%. *Presenter encouraged questions/participation*: agree, 83.6%; neutral, 14.9%; disagree, 1.5%. *Presenter built rapport with participants*: agree, 85%; neutral, 13.4%; no response, 1.5%. [D] Encourage speaking loudly. [D] Interaction with other regions is interesting. | *Presenter stimulated critical thinking*: agree, 73.2%; neutral, 19.4%; disagree, 6%; no response, 1.5%. *Information revealed was not available elsewhere*: agree, 34.3%; neutral, 31.3%; disagree, 31.4%; no response, 3.0%. *Found useful tips for practice*: agree, 53.8%; neutral, 28.4%; disagree, 13.5%; no response, 3.5%. *Caused reflection on practice*: agree, 70.1%; neutral, 11.9%; disagreed, 16.4%; no response, 1.5%. *Will change current practice*: yes, 25.4%; no, 59.7%; not sure, 6.0%; problems, 6.0%; no response, 6.0%. [D] Excellent platform for continuing education. [D] Need more didactic teaching. | *Presenter visible*: agree, 92.6%; neutral, 4.5%; disagree, 3%. *Slides/visual aids visible*: agree, 55.5%, neutral, 17.9%; disagree, 17.9%; no response, 9.0%. *Presenter audible*: agree, 67.1%; neutral, 25.4%; disagree, 6%; no response, 1.5%. | |

*(Continued)*

**Table 2.** (Continued)

| Study | Overall value and preference | Organization | Case Discussions and Decision-making | Teamwork and Communication | Education and Training | Technology | Patient-centered aspect |
|---|---|---|---|---|---|---|---|
| Olver IN *et al.* 2000 | *Overall satisfaction*: will use the VMDTM, 95%; will not use the VMDTM, 5%. | *Impact on workload*: same, 50%; better, 25%; worse, 25%. *Preferred format for documenting meeting*: written, 60%; video, 5%; video and written, 25%; no opinion, 10%. [D] Lack of reimbursement is a barrier. | *Useful for peer review*: useful, 55%; not very useful, 20%; never used/not applicable, 25%. | [D] Impersonal nature of does not portray the clinical ability of other physicians. [D] Isolated physicians felt better supported. | *Useful for education*: useful, 19/20; never used/not applicable, 1. *Impacted personal practice*\*: yes, 50%. [D] \*Reasons: replacement of phone consultations, impact on management without patient travel, better peer review, better treatment plan. | [D] Technological problems exist: poor image quality, moving objects, break-downs and display issues with pathology/ radiology images. *Concerns about patient confidentiality*: concerning, 75%; not very, 10%; no opinion, 15%. | *Patients should be present*: no, 50%; yes, 35%; no opinion, 15%. [D] Unable to cross-examine patients is problematic. |
| Hunter *et al.* 1999 | | *Day of meeting is suitable*: yes, 100%. *Time of meeting is suitable*: yes, 97%; no, 3%. *Time for case discussion*: excellent/ good, 95%; fair, 5%. *Volume of cases is adequate*: yes, 100%. | *Contribution of radiology*: major, 46%; minor 32%; did not contribute, 8%. *Contribution of pathology*: major, 87%, minor, 13%. *Contribution of surgical oncology*: major, 84%, minor, 16%. *Contribution of medical oncology*: major, 82%; minor, 16%; did not contribute, 2%. *Contribution of radiation oncology*: major, 63%, minor, 24%; did not contribute, 13%. | | | *Quality of audio*: excellent/good, 79%; fair, 21%. *Quality of participants' video*: NA, 100%. *Quality of mammograms*: excellent/good, 89%, NA, 10%. *Quality of pathology slides*: excellent/good, 95%, NA, 3%. | |
| | | *Day of meeting is suitable*: yes, 95%; no, 5%. *Time of meeting is suitable*: yes, 85%; no, 15%. *Time for case discussion*: excellent/ good, 95.3%; fair, 4.8%. *Volume of cases is adequate*: yes, 81%; no, 19%. | *Contribution of radiology*: major, 100%. *Contribution of pathology*: major, 95.5%, minor, 4.5%. *Contribution of surgical oncology*: major, 95.5%, minor, 4.5%. *Contribution of medical oncology*: major, 90.9%; minor, 4.5%; did not contribute, 4.5%. *Contribution of radiation oncology*: major, 95.5%, minor, 4.5%. | | | *Quality of audio*: excellent/good, 100%. *Quality of participants' video*: excellent/good, 100%. *Quality of mammograms*: excellent/good, 95%, fair, 5%. *Quality of pathology slides*: excellent/good, 95%, fair, 5%. | |

[D] Indicates that the finding was qualitatively cited, either through the use of an interview, an anthropological analysis (Delaney *et al.*), or as a summary from the study's data which was not otherwise quantitatively stated in the relevant paper.

[C] Indicates that the finding compares VMDTM with IMDTMs. Where numbers are quoted, VMDTMs have been compared with IMDTMs i.e. VMDTM vs. IMDTM.

**Abbreviations**: CME, continuing medical education; IMDTM, in-person multidisciplinary team meeting; MODe, metric of decision-making; MOT, meeting observational tool; VMDTM, virtual multidisciplinary team meeting.

### 3.2.A Organization

A total of 21 studies evaluated one or more organizational aspects of the VTB [7, 17, 19, 21, 23, 25, 27, 28, 34, 35, 38, 39, 42–44, 46, 49]. Overall, VMDTMs may be equally organized as IMDTMs [27] if not more [17, 23].

**Attendance.** The foremost benefit of the virtual nature of VMDTMs has been the removed need to schedule rooms for the meeting, thereby facilitating convenient remote access for participants [19, 43]. This benefits the meeting itself by impacting clinician attendance as, since the advent of the virtual format, participant attendance in board meetings has reportedly increased [19, 43, 44]. However, one study, that assessed responses from a national VMDTM, reported sub-optimal attendance in the national VMDTM owing to irregular meeting times and a lack of resources [49].

**Scheduling and time allotment.** Three studies evaluated perceptions regarding the schedule of VMDTMs. Participants were satisfied with the day and time of the meeting across these studies [5, 38, 41]. Similarly, participants were also content with the frequency of the meetings [28, 41]. Several studies evaluated the time allotted for case discussions which was reportedly adequate across many studies [5, 7, 28, 29, 35, 38].

**Preparation for the meeting.** Preparing for the meeting is an important aspect of VMDTMs. One study reported that most radiologists spent 1–4 hours per week preparing for the VMDTM [19]. This aspect was deemed time-consuming in two studies [46, 49], but was adequate for participants in another study [28]. However, overall VMDTMs possess a considerable time-saving benefit [17, 39, 43]. In comparison to IMDTMs, Olver *et al.* reported that VMDTMs do not affect physicians' workload [30]. Similarly, Mohamedbhai *et al.* reported that the time efficiency in VMDTMs is comparable to that in IMDTMs [23].

**Chairing the meeting.** Opinions regarding chairing the meeting varied. A few participants in the study by Bonanno N *et al.* found it easier because of less people spoke simultaneously [19]. However, when compared to IMDTMs, overlap during conversations may exist, and may pose a considerable hurdle for trainees [21]. Although a majority of the study by Mohamedbhai *et al.* reported that chairing is difficult in VMDTMs compared to IMDTMs [23], participants found leadership in VMDTMs adequate [17, 39, 49].

**Clarity of roles.** Rosell *et al.* evaluated participants' perception of their roles during VMDTM meetings, and although a majority agreed that their roles are clear [42], there is still a need to further clarify roles better in VMDTMs as evaluated in a subsequent study by the same author [49].

**Meeting minutes.** Recording meeting outcomes may be more difficult when compared to IMDTMs [23]. However, clear guidelines about recording meeting minutes can help circumvent this hurdle [42]. One study evaluated the importance of meeting minutes and reported that participants almost always used them to stay updated [28]. In another study, the written format was the most popular method for documenting meeting notes [5].

### 3.2.B Case discussion and decision-making

A total of 27 studies evaluated perceptions regarding one or more aspects related to case discussions and decision-making during VMDTMs [5, 7, 17–21, 23, 24, 27–30, 33–35, 38–42, 45–50]. Various differing aspects were evaluated across studies.

**Usefulness in discussing complex cases.** The most frequently discussed sub-theme of VMDTMs was their usefulness in discussing complex cases [18, 45, 46, 49]. According to one study, discussing routine cases, which have established guidelines, is counterproductive to the meeting and can even irritate participants [45]. Participants were satisfied with the selection of cases for discussion in the VMDTMs in two studies [39, 46]. VMDTMs also provide support

to physicians in making difficult clinical decisions [39] and allow peer review of management plans [30].

**Case discussion.** It is well-established that discussions in VMDTMs provide case-specific information that impacts patient management [18, 24, 28, 38, 40, 47]. Discussing cases in VMDTMs helps reduce redundancies in patient management [40]. According to multiple studies, the clinical decision process [27] and the comprehensiveness of recommendations [7, 19, 21] are almost similar between VMDTMs and IMDTMs.

Presentations and meetings were smooth during VTBs and rated highly in two studies [29, 35]. In the same studies, participants were well satisfied with the knowledge of their fellow participants [35] and that of the presenter [29].

**Decision-making.** Two studies reported that consensus was frequently reached in VMDTMs [20, 35], and, according to one study, there is no difference between IMDTMs and VMDTMs in reaching a consensus [46]. However, one study reported that the level of consensus was slightly lower for VMDTMs compared to in-person meetings [48]. Amongst the evaluated studies, we saw no difference between VMDTMs and IMDTMs in decision-making [7, 17, 23]. In one study, decision-making in VMDTMs scored slightly lower than in IMDTMs, but scores were high for both formats [47].

As reported in one study, VMDTMs made use of information from clinical studies during VMDTMs in comparison to IMDTMs [40]. Thus, the quality of information used to manage cases was better in VMDTMs when compared to IMDTMs, likely because data from clinical studies was used [40]. Similarly, another study that conducted an anthropological analysis, saw that participants used scientific evidence to support decision-making in VMDTMs, compared to multiple ways of knowing in IMDTMs [50].

**Availability of information and specialties.** Availability of background patient information was reportedly adequate for a productive discussion in VMDTMs in multiple studies [41, 42, 46, 47]. Three studies evaluated the contribution and role of pathology and radiology in VMDTMs [5, 41, 50]. In two studies, a majority of respondents rated the contribution of both fields to the meeting highly [5, 50]. Savage *et al.* commented on the quality of radiology and pathology information, rating it highly [41].

Participants were satisfied with the number of specialties on board during VMDTMs [7]. Delaney *et al.* reported that participants believed that attendance significantly aids case management [50]. Schroeder *et al.* reported that VMDTM participants were more experienced and better qualified when compared to IMDTMs [40].

## 3.2.C Teamwork and communication

A total of 20 studies assessed perceptions regarding teamwork and communication [7, 17–19, 21–23, 27, 29, 30, 35, 38, 40–43, 47–50].

**Teamwork.** The level of satisfaction of participants with the level of teamwork in VMDTMs was high in many studies [7, 19, 22, 29, 35, 41, 42, 48]. Participants felt involved as they were called upon to contribute [42]. They also expressed satisfaction with the equitable distribution of discussion among specialties [7, 35, 48]. However, in one study the participants felt that the virtual nature negatively impacts the quality of teamwork during VTB meetings [23]. Up to 44% felt that teamwork is worse in VMDTMs, compared to IMDTMs [23].

Participants in two studies reported that interaction between specialties in VMDTMs was adequate [7, 41]. In a recent study, engagement of specialties was reportedly better in VMDTMs compared to IMDTMs [17]. More than 70% of respondents in one study reported that discussion was also adequately shared between specialists, and more than 88% agreed that VMDTMs did not limit the number of specialties that can be present on board [7].

Furthermore, VMDTMs support isolated physicians who otherwise would not be able to access a multidisciplinary discussion forum [30].

**Communication.**   VMDTMs allow better communication among physically distant healthcare providers [40]. Up to 70% of respondents in one study opined that VMDTMs improve coordination between hospitals and private clinics [40]. When seeking recommendations from various specialties, VMDTMs provide a comparable level of ease as IMDTMs [21]. VMDTMs also improve communication between specialties [18, 19, 38], but in comparison to IMDTMs, it may be worse [23]. While a significant majority of participants in another study noted the similarity in communication between VMDTMs and IMDTMs, up to 42% in the same study expressed a preference for IMDTMs due to enhanced communication quality [27].

Amongst the biggest impacts of the virtual format is the inability to be physically present with other members. The impersonal nature of the virtual format prevents participants from catching on non-verbal cues which can lead to misunderstandings [17, 19, 43, 50]. According to a study, the impersonal nature also hinders the ability to assess the ability of other physicians [30].

**Networking.**   VMDTMs are a good platform for physicians to meet other specialties [18]. VMDTMs also bring healthcare providers together over geographically distant sites fostering valuable opportunities for professional networking [49]. However, maintaining these working relationships with members is difficult [19], which is seen in particular with new members of the team [17, 19, 43]. When compared to IMDTMs, networking with other specialties is difficult [21] and interpersonal relationships deteriorate [23] in VMDTMs.

Interaction with the case presenter was assessed in two studies and was rated highly [29, 35]. A considerable majority agreed that the presenter encouraged questions and input from the VTB team [29, 35].

### 3.2.D Education/training

Of the studies included in this study, a total of 20 studies evaluated the educational importance of VMDTMs [17, 18, 20, 21, 23, 26–30, 33–37, 39–42, 46, 49].

**Educational value.**   Almost all studies highlighted the usefulness of VMDTMs for their educational value [17, 18, 20, 27–30, 33–35, 39, 41, 42, 46, 49]. In only one study a majority of participants (48%) reported that training in VMDTMs was worse than in IMDTMs [23]. Despite this, approximately 44% of respondents within the same study said that the training is on par or superior to that in ITBs [23].

It is the extensive scientific discussion of cases and the multidisciplinary nature of VMDTMs that provide advanced training to its members [39]. In doing so, VMDTM discussions encourage critical thinking [29, 35]. Many studies reported that attending VMDTMs helped improve individual [27, 33, 42, 49] and team competency [42, 49]. In two studies, more than 80% of respondents reported that they perceived an increase in knowledge after attending VMDTMs [33, 40].

The periodical and continued occurrence of VMDTMs provides a consistent means of medical education to attending participants [28, 29]. The educational nature of VMDTMs is appropriate for their consideration as CMEs [18, 33]. In one study, participants received CME credits for attending VMDTMs, and getting CMEs was equally easy between VMDTMs and IMDTMs for participants [21].

VMDTMs at the national level allow physicians from low-volume centers to see cases that they would not typically encounter in their clinical practice [49]. Similarly, low-volume centers are exposed to numerous cases that would otherwise be outside their purview, and this helps increase physician experience [49]. Furthermore, the presence of varying experts provides

information that, at times, may not be available elsewhere in other educational sources [29, 35]. Where time consumed reviewing literature for the meeting can be viewed as a potential drawback, it, too, inadvertently serves to increase clinical knowledge [46].

**Impact on clinical practice.** VMDTMs are beneficial for clinical practice [39] as case discussions allow participants to reflect on their practice [35]. Two studies saw that VMDTMs offered clinical tips to attending physicians [29, 35], which in turn impacted their practice [29, 30, 35, 37]. Furthermore, VMDTMs make physicians confident about their management plans [26, 36].

## 3.2.E Technology

A total of 23 studies evaluated the technological aspect of VMDTMs [5, 7, 17, 19, 21–23, 27–30, 33–36, 39, 42–44, 46, 47, 49, 50]. Overall satisfaction with technology in VMDTMs was evaluated in four studies where a majority of participants were satisfied across all four studies [23, 36, 42, 46]. Around 77–88% of respondents in three studies said that technological problems barely affected VMDTMs [23, 36, 46]. Despite high overall satisfaction with technology in VMDTMs, technological issues exist and can be a significant hurdle for VMDTMs [28, 43, 44]. Furthermore, the switch to a virtual format can cause participants short-term difficulties, which subside once participants become familiar with the format [22].

**Accessing patient data and other resources.** Accessing imaging data was smooth for 71–86% of respondents in two studies [7, 19]. VMDTMs do not limit access to patient information [7, 47]. In one study, VMDTMs scored higher than IMDTMs in the domain of the provision of physical resources for the meeting [47]. Sending pathology samples for review in VMDTMs is also easy, as reported in one study [28]. However, transferring case-related information may be time-consuming [49].

**Imaging quality.** The quality of case-related imaging has reportedly been good over the years, even as far back as the year 1999 [5, 19, 21, 23, 27, 41]. When compared to IMDTMs, up to 70% of participants in three studies found image quality equal to or even better than IMDTMs [19, 23, 27]. Good image quality translates to easier image review as reported in a study where trainees found it significantly easier than other participants (e.g. staff) to review case images in VMDTMs when compared to IMDTMs [21].

**Audio and video quality.** Audio and video quality was the most commonly evaluated technological feature of VMDTMs. Participants were satisfied or highly satisfied with the audio and video quality across all studies that evaluated these features [5, 22, 29, 34, 35, 50]. In three studies, video and audio quality was rated very highly [5, 34, 35]. In the remaining three studies, although not in the majority, a considerable proportion of respondents were unsure about or not satisfied with the video and audio quality [22, 29, 50].

**Data security.** Data security in VMDTMs is equal to IMDTMs, as reported by up to 68–80% of participants across two studies [23, 27]. This is unlike the results from an older study in 2000 where patient confidentiality in VMDTMs was deemed "concerning" [30].

**Technical support.** Only one study evaluated technical support for VMDTMs, where it was rated adequate by participants [39]. When missing, the lack of technical support can be a significant barrier for participants [17, 19, 39].

**Technological barriers.** Improving technological structure is one of the most important changes to improve VMDTMs [43, 44], as shortcomings in this domain may hinder participation [7]. The major technological challenge was connectivity across most studies [17, 19, 21, 30, 43, 46, 49]. Other issues evaluated in the included studies were viewing images [19, 30], viewing presentations [21], issues with audio [21], and lack of technical support [19].

### 3.2.F Patient-centered aspect

A total of 12 studies commented on patient-related aspects of VMDTMs [17, 18, 20, 23, 27, 30, 36, 37, 41, 42, 46, 49].

**Opportunity for clinical trial recruitment.**   The most commonly discussed feature of VMDTMs was their potential to facilitate the recruitment of patients for clinical trials [17, 18, 37, 41, 46, 49]. Around 44–77% of participants across three studies agreed that VMDTMs allow clinical trial recruitment [18, 37, 46]. However, up to 40% of participants in two studies were unsure about this feature [18, 41]. In one of these studies, the majority (38%) disagreed [41]. In another study, participants stated that there was limited focus on clinical trial recruitment, and instead highlighted that time dedicated to research protocols could generate new research initiatives [49]. In comparison to IMDTMs, VMDTMs offer no further benefit for clinical trial recruitment [17].

**Consideration of patient comorbidity and perspectives.**   Only the studies by Rosell *et al.* commented on the consideration of patient comorbidity and patient views during the VMDTMs [42, 49]. Although more than 70% of participants in the study in the year 2019 agreed that comorbidity and perspectives are considered [42], the study in the year 2020 reported limited consideration of patients' comorbidities [49].

**Benefits for patients.**   While VMDTMs allow physicians to generate meticulous management plans, they also positively impact patients. Evidence suggests that VMDTMs boost patients' confidence in the management plan [36]. Furthermore, VMDTMs also help provide patients with information about clinical trials and guidelines [18]. Moreover, where VMDTMs help physicians connect over geographical distances, they offer an alternative solution to patients who may be reticent about seeking care at a different medical care facility [46].

## 4 Discussion

To the best of our knowledge, this is the first overall review to assess the feasibility and acceptance of VMDTMs by assessing the perspectives of meeting participants. Across all included studies, we recognized and assessed six core components of a VMDTM which have been previously described for MDTMs [11]. We observed a high level of satisfaction with the overall dynamics of the VMDTM across many studies [7, 19, 21, 22, 24, 26, 27, 29–33, 35–38, 40, 46, 47, 49, 50]. However, preference was reserved for a hybrid model of MDTMs [17, 19, 23, 43–45].

Despite a high level of acceptance for VMDTMs, the format was limited in certain aspects as described in **Table 3**. Lack of reimbursement was a barrier to attendance for isolated or community-based physicians. This is supported by prior literature where the equipment cost VMDTMs defer physicians from attending VMDTMs [12]. It is important to highlight that VMDTMs may be more cost-effective than IMDTMs in the long term [51, 52].

Well-functioning technology is crucial to a VMDTM and its inadequacy is a major limitation that could make VMDTMs unsustainable [7, 19, 30, 43]. The major technological issue highlighted in our studies was poor connectivity or limited bandwidth. Limited bandwidth prevents the proper functioning of audio and video components simultaneously, so, a bandwidth of 2 Mbps has been previously recommended [12].

Although multiple studies reported that a majority of participants agree that VMDTMs allow recruitment for clinical trials, not all participants were convinced about this feature [18, 37, 46]. A considerable portion of respondents were neutral to support this feature [18, 37, 46], and two studies disagreed with this capability [41, 49]. The limited number of studies and the conflicting data put this feature in doubt.

MDTMs have proven their merit in providing and impacting management decisions [12, 53]. The same was seen in our review [18, 24, 28, 38, 40, 47]. The most frequently reported

**Table 3. Limitations for VMDTMs reported across included studies.**

| Limitations | Comment |
|---|---|
| **Organization** | |
| Lack of reimbursement | For attending physicians, the lack of reimbursement for presenting cases in VMDTMs was a barrier to attendance, as described in two studies fourteen years apart [30, 46]. In the study by Shea *et al.*, this was an important barrier for community-based physicians [46]. |
| Scheduling | Clash with other commitments was reported as a scheduling issue in two studies [28, 38]. Like other meetings, it is necessary to consider VMDTMs consistently like any other meeting in order to not let the virtual ease of attending the format create scheduling problems. |
| Preparing cases is time-consuming | Although participants in two studies reported that preparing cases was time-consuming [46, 49], participants in other studies reported that they had adequate time [19, 28]. Furthermore, VMDTMs were deemed time-efficient in other studies [17, 23, 43]. |
| **Case Discussion and Decision-Making** | |
| Varying level of preparedness of participants | Rosell *et al.* reported that discussion quality is limited if participants are unprepared for the meeting [49]. This necessitates that attending participants should prepare for the cases well before the meeting. |
| Lower commitment for referred cases | Rosell *et al.* reported that participants in meetings showed lower dedication towards referred cases [49]. |
| Resolution within one specialty | Delaney *et al.* reported the decision-making was majorly reserved within one specialty [50]. This is in comparison to multiple other studies that reported adequately shared decision-making [7, 17, 22, 23, 35, 47, 48]. |
| **Teamwork and Communication** | |
| Loss of non-verbal cues cause misunderstandings | Misunderstandings due to unacknowledged uncertainties was reported by multiple studies [19, 49, 50]. A solution for this drawback was reported by Ali *et al.* who recommended making use of hands-up alerts present within the software [17]. |
| Impersonal nature affects meetings | According to Delaney *et al.* the impersonal nature of the virtual format, inevitably eliminated a considerable amount of social aspects such as the ability to joke with colleagues, open conversation, and the equality between participating centers [50]. |
| **Education/Training** | |
| Training | Only participants in the study by Mohamedbhai *et al.* believed that training was poor in VMDTMs compared to the in-person format [23]. In stark contrast, all the other studies noted that VMDTMs are beneficial and good at offering clinical education and training [17, 18, 20, 27–30, 33–35, 39, 41, 42, 46, 49]. |
| **Technology** | |
| Connectivity | Internet connectivity and limited bandwidth was the most commonly described technical limitation [17, 19, 21, 30, 43, 46, 49]. It was deemed an important barrier to VMDTMs. The implications of poor connectivity were described as: the inability to see other participants [49], misunderstandings between participants [49], compromised case-discussion [43, 46], and meeting delays [44]. |
| Audio/video quality | Audio/video quality was satisfactory across all studies that evaluated these aspects of VMDTMs [5, 22, 29, 34, 35, 50]. The reported consistency of adequate audio/video quality was reported as far back as 1999 in the study by Hunter *et al.* [5]. |
| Image quality | In 2000, Olver *et al.*, reported concerning image quality [30]. However, subsequent studies in recent years have reported adequate or an even better image quality compared to IMDTMs [17, 19, 21, 23, 27]. |
| Logistics of handling patient data | Olver *et al.* reported, in the year 2000, that displaying radiology and pathology information was difficult [30]. In the year 2020, Rosell *et al.* considered the process of transferring radiology slides time-consuming but did not comment on the level of difficulty [49]. In a 2013 study by Abu Arja *et al.*, pathology slides were easy to transfer, but no comment was made on whether the process was time-consuming or not [28]. |

(*Continued*)

**Table 3.** (Continued)

| Limitations | Comment |
|---|---|
| Lack of technical support | The lack of technical support was a barrier to attendance in the study by Bonanno *et al.* [19]. Similarly, the study by Ali *et al.* cited that the availability of technical support would be an important improvement for VMDTMs [17]. In the study by Chekerov *et al.*, technical support was deemed adequate [39]. |
| **Patient-related aspect** | |
| Clinical trial recruitment | A majority of VMDTM participants agreed that VMDTMs are valuable in identifying patients for clinical trials. However, a considerable portion of respondents, in the same studies, were unsure or disagreed, which put this aspect of VMDTMs into doubt [18, 41] |
| Concerning patient confidentiality | Olver *et al.* reported concerns with patient confidentiality in the year 2000 [30]. However, confidentiality over the years has improved and is comparable to IMDTMs [17, 23, 27], in recent years. |
| Limited consideration of patient comorbidity and perspectives | Only the studies by Rosell *et al.* evaluated consideration of patient perspectives or comorbidity in VMDTMs [42, 49]. Thematic analysis of semi-structured interviews reported a limited consideration in comparison to 71–79% of participants agreeing that patient views are taken into account during VMDTMs [42, 49]. |

**Abbreviations**: IMDTM, in-person multidisciplinary team meeting; VMDTM, virtual multidisciplinary team meeting.

Note: This table lists down all limitations of VMDTMs as they were identified amongst the included studies. The table offers an explanation, solution, or a general comment on each limitation which is supported by data from existing literature.

feature was their usefulness in discussing complex cases [18, 45, 46, 49]. The study by van Huizen *et al.* reported that discussing routine cases was detrimental to the concept of a VMDTM [45], which was similar to previously reported literature where discussing all cases was considered inefficient to the concept of an MDTM [54–56].

VMDTMs were not limited in their level of peer review [30] their meeting depth [7, 19], or decision-making [20, 35]. Schroeder *et al.* reported that VMDTMs incorporated data from clinical studies to make management decisions [40]. This was supported by Delaney *et al.* who saw that scientific evidence was used in VMDTMs compared to multiple ways of knowing in ITBs [50]. This feature of VMDTMs makes them reliable and more clinically oriented than IMDTMs.

The multidisciplinary nature of VMDTMs makes them an excellent learning platform. All the studies that evaluated the educational potential of VMDTMs, saw that almost all participants saw VMDTMs as fit for one or more educational aspects [17, 18, 20, 27–30, 33–35, 39, 41, 42, 46, 49]. So much so, that their consideration for CME credits was advocated for. Multidisciplinary teams have been CME accredited previously [57] and consideration of VMDTMs may also be appropriate.

Furthermore, by overcoming geographical distances, VMDTMs allow remote centers to witness more cases than they normally would [49]. Nation-wide referrals help physicians see cases from across the country that they normally would not see in their locations [49]. This unparalleled benefit over IMDTMs highlights complex and challenging cases that are rare to encounter for low-volume centers.

There are multiple limitations to this study. The varying methods of reporting data limit the uniformity of data throughout the included studies. In addition, this study is limited to the subjective opinion and acceptance of the virtual format.

Despite these limitations, this study provides a strong summary that is based on several VMDTM members and their experiences. Therefore, our study forms the basis for future modifications and gives a genuine insight into the workings of a contemporary VMDTM.

## 5 Conclusion

Where VMDTMs connect physicians across borders, offer clinical support to isolated physicians, and offer solutions to clinical questions that do not have answers elsewhere, they are limited in certain aspects. The impersonal nature is detrimental to a candid atmosphere. More importantly, when the fundamental technological pillar, that allows the virtual format to exist, is weak, the sustainability of VMDTMs becomes doubtful. Yet, institutions that meet the necessary criterion may experience a smooth, and in some cases, a VMDTM better than its in-person counterpart. The popularity of VMDTMs has grown tremendously throughout the pandemic further focused assessments of VMDTMs' core components must be made to better streamline the VMDTM experience.

## Supporting information

**S1 File. PRISMA ScR checklist.** This file contains the PRISMA ScR checklist.
(PDF)

**S2 File. Search string(s).** This file contains the individual search strings used for each database to perform the literature search for this study.
(PDF)

**S3 File. List of excluded studies.** This file contains the list of studies that were excluded, along with their reasons for exclusion, after full-text review.
(PDF)

**S4 File. Composition and percentage responses.** This file highlights further characteristics of included studies, their populations, and VMDTM composition.
(PDF)

**S5 File. Distribution of included studies.** This file shows the country/region-wise distribution of populations in included studies.
(PDF)

**S6 File. Additional findings.** This file shows additional findings in included studies.
(PDF)

## Author Contributions

**Conceptualization:** Muhammad Abdul Rehman.

**Data curation:** Muhammad Abdul Rehman, Unaiza Naeem, Anooja Rani, Umm E. Salma Shabbar Banatwala, Afia Salman, Muhammad Abdullah Khalid, Areeba Ikram.

**Methodology:** Muhammad Abdul Rehman, Unaiza Naeem.

**Supervision:** Muhammad Abdul Rehman, Unaiza Naeem.

**Validation:** Muhammad Abdul Rehman.

**Writing – original draft:** Muhammad Abdul Rehman, Unaiza Naeem, Anooja Rani, Umm E. Salma Shabbar Banatwala, Afia Salman, Muhammad Abdullah Khalid, Erfa Tahir.

**Writing – review & editing:** Muhammad Abdul Rehman, Erfa Tahir.

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
