## [Decision Letter · Decision Letter 0]

31 Oct 2023

PONE-D-23-30802How Well Does the Virtual Format of Oncology Multidisciplinary Team Meetings Work? An Assessment of Participants’ Perspectives and Limitations: a Scoping ReviewPLOS ONE

Dear Dr. Abdul Rehman,

Thank you for submitting your manuscript to PLOS ONE. After careful consideration, we feel that it has merit but does not fully meet PLOS ONE’s publication criteria as it currently stands. Therefore, we invite you to submit a revised version of the manuscript that addresses the points raised during the review process.

We look forward to receiving your revised manuscript.

Kind regards,

Suzanne Rose

Academic Editor

PLOS ONE

Journal Requirements:

"All authors of this review work with, or have worked with, the not-for-profit organization, “Tumor Board Establishment Facilitation Forum (TEFF)”. TEFF is a student-run organization that helps establish, and offers access to, multidisciplinary tumor boards at the tertiary care hospital, Dr. Ruth Pfau Civil Hospital in Karachi, Pakistan."

Additional Editor Comments:

Thank you so much for this timely submission on a relevant topic. Kindly address all reviewer comments to fully re-submit the paper for full consideration to the journal. 

Reviewers' comments:

Reviewer's Responses to Questions

**Comments to the Author**

1. Is the manuscript technically sound, and do the data support the conclusions?

Reviewer #1: Yes

Reviewer #2: Yes

2. Has the statistical analysis been performed appropriately and rigorously? 

Reviewer #1: Yes

Reviewer #2: N/A

3. Have the authors made all data underlying the findings in their manuscript fully available?

Reviewer #1: Yes

Reviewer #2: Yes

4. Is the manuscript presented in an intelligible fashion and written in standard English?

Reviewer #1: Yes

Reviewer #2: Yes

5. Review Comments to the Author

Reviewer #1: Authors report a scoping review on the virtual format of oncology multidisciplinary team meetings focusing in particular on the assessment of participants’ perspectives and limitations. The paper is interesting since it analysed the question in all cancer settings. In my opinion its may ben accepted in the present form.

Reviewer #2: Thank you for the opportunity to review your work. Overall, you have completed a well-written, PRISMA-compliant Scoping review.

I do have several minor comments that, when addressed, will hopefully strengthen your work:

Line 47 - you set up VMDTM as a "promising solution" however, you hadn't introduced a problem yet.

Line 60 - define IMDTMs before use.

Line 62 - this comment likely falls outside of the context of your scoping review; however, I'm curious if there were perception changes pre-during-post pandemic...

Line 274 contradicts 278. VMDTMs are good to meet other specialities and then it says networking with other specialities is difficult and relationships deteriorate. In the attendence section (165) you did a better job with transition... some reports state its good, however, others its bad. Use a transition in the paragraph, please.

Line 382 - first overall review or first scoping review?

Line 435 - "this allows this" awkward

PRISMA Flow:

make records identified from Databases N=2617 (the individual databases can be "little n")

I have slight concern that you distrinction of new studies vs reports of new included studies incorrectly. seems like the new studies are from the "left side" and the reports are from the "right side" It could be correct, but it's a big coincidence when looking at left vs. right. Check this out, it's the best PRISMA clarifications that I've come across: DOI 10.5195/jmla.2022.1449

Supplement 4 - the final row (breast VMDTM) is missing author information.

Supplement 6 - the final row (Lighting quality 100%) is missing author information.

6. PLOS authors have the option to publish the peer review history of their article (what does this mean?). If published, this will include your full peer review and any attached files.

Reviewer #1: No

Reviewer #2: **Yes: **Eric C. Nemec

---

## [Author Response · Author response to Decision Letter 0]

1 Nov 2023

Manuscript ID PONE-D-23-30802

Title How well does the virtual format of oncology multidisciplinary team meetings work? An assessment of participants’ perspectives and limitations: a scoping review

I would like to thank both reviewers for taking the time out to read our manuscript and for their interest and their comments.

Reviewer 1 had no comments for our paper. We have addressed all the comments raised by Reviewer 2. 

Comment: Line 47 - you set up VMDTM as a "promising solution" however, you hadn't introduced a problem yet.

Response: The sentence has been edited to highlight that the solution is for geographical barriers, as was conveyed by the latter half of the sentence. I thank the reviewer for raising this comment. 

Comment: Line 60 - define IMDTMs before use.

Response: Thank you for this comment. The abbreviation has been defined. 

Comment: Line 62 - this comment likely falls outside of the context of your scoping review; however, I'm curious if there were perception changes pre-during-post pandemic…

Response: This is an interesting comment. I agree with the reviewer that evaluating a change in perception about VMDTMs before and after the COVID pandemic, is actually not the focus of this study and it also does not sit well with the nature of this review. It would perhaps form the basis for a new systematic review which specifically focuses on this particular question. 

Despite this, I would like to point out that in accordance with the scoping nature of our review, we have, indeed made the distinction for studies that were published during or after the COVID-19 pandemic (n=10). This has been stated in Table 1 and the results section (Line 140). 

Comment: Line 274 contradicts 278. VMDTMs are good to meet other specialities and then it says networking with other specialities is difficult and relationships deteriorate. In the attendence section (165) you did a better job with transition... some reports state its good, however, others its bad. Use a transition in the paragraph, please.

Response: I would like to respond to this comment by highlighting the sentences in the manuscript. We state that “VMDTM are good to meet other specialties…”, which is followed by "However, maintaining these working relationships with members is difficult”. Although VMDTMs help physicians meet physicians from other specialties, it does not translate into long-lasting professional relationships which is why “When compared to IMDTMs, networking with other specialties is difficult and interpersonal relationships deteriorate”. 

I welcome the reviewer’s comment and the opportunity to respond. But as stated by the quotations above, there are two things to consider: (1) That the distinction between “meeting” and “maintaining relationships” is to be considered, which is conveyed by the current text; (2) that the second sentence is actually comparing VMDTMs to IMDTMs and is not about VMDTMs entirely. 

Comment: Line 382 - first overall review or first scoping review?

Response: This is the first overall review that assesses VMDTM feasibility through the eyes of VMDTM participants. I have edited the sentence to highlight the distinction as recommended by the reviewer. 

Comment: Line 435 - "this allows this" awkward

Response: We have restructured the sentence from “This allows this paper to form the basis…” to “Therefore, our study forms the basis…”

Comment: PRISMA Flow:make records identified from Databases N=2617 (the individual databases can be "little n") I have slight concern that you distrinction of new studies vs reports of new included studies incorrectly. seems like the new studies are from the "left side" and the reports are from the "right side" It could be correct, but it's a big coincidence when looking at left vs. right. Check this out, it's the best PRISMA clarifications that I've come across: DOI 10.5195/jmla.2022.1449

Response: I thank you for this particular comment and will elaborate both aspects of the question. The lowercase “n” is used to denote the number of studies throughout the flowchart. And so, it denotes the number of studies at “Databases n=2617” and not the number of databases. Since the essence of the variable n has not changed, it seems best to quote it as such. I agree with the reviewer that an uppercase “N” should be used, only if N represented something else than n. The lowercase n in the figure is also in-line with the official PRISMA 2020 flowchart which can be found here (http://prisma-statement.org/prismastatement/flowdiagram.aspx). Similarly, the distinction of new included studies and the reports of new included studies is also in-line with the PRISMA 2020 flowchart. We kept our citation search in the second column because it was conducted as a secondary search. This is in-line with the article quoted by the reviewer - which was indeed an interesting read.

Comment: Supplement 4 - the final row (breast VMDTM) is missing author information.

Response: The final row belongs to the study by Hunter et al. Therefore, I have merged the cells to indicate as such.

Comment: Supplement 6 - the final row (Lighting quality 100%) is missing author information.

Response: The final row belongs to the study by Hunter et al. Therefore, I have merged the cells to indicate as such.

---

## [Editor Report · Decision Letter 1]

6 Nov 2023

How well does the virtual format of oncology multidisciplinary team meetings work? An assessment of participants’ perspectives and limitations: a scoping review

PONE-D-23-30802R1

Dear Dr. Rehman,

We’re pleased to inform you that your manuscript has been judged scientifically suitable for publication and will be formally accepted for publication once it meets all outstanding technical requirements.

Kind regards,

Suzanne Rose

Academic Editor

PLOS ONE

Additional Editor Comments (optional):

Thank you for your thoughtful revisions to the review comments. Congratulations on the acceptance of your article to the journal!

Reviewers' comments:

No additional reviewer comments were requested due to the minor revision request.

---

## [Editor Report · Acceptance letter]

9 Nov 2023

PONE-D-23-30802R1 

How well does the virtual format of oncology multidisciplinary team meetings work? An assessment of participants’ perspectives and limitations: a scoping review 

Dear Dr. Abdul Rehman:

I'm pleased to inform you that your manuscript has been deemed suitable for publication in PLOS ONE. Congratulations! Your manuscript is now with our production department. 

Kind regards, 

on behalf of

Dr. Suzanne Rose 

Academic Editor

PLOS ONE